# Common alleles of *CMT2* and *NRPE1* are major determinants of CHH methylation variation in *Arabidopsis thaliana*

Eriko Sasaki[1], Taiji Kawakatsu[2,3,4], Joseph R. Ecker[2,3,5], Magnus Nordborg[1]*

**1** Gregor Mendel Institute of Molecular Plant Biology, Austrian Academy of Sciences, Vienna Biocenter, Vienna, Austria, **2** Plant Biology Laboratory, Salk Institute for Biological Studies, La Jolla, California, United States of America, **3** Genomic Analysis Laboratory, Salk Institute for Biological Studies, La Jolla, California, United States of America, **4** Institute of Agrobiological Sciences, National Agriculture and Food Research Organization. Tsukuba, Ibaraki, Japan, **5** Howard Hughes Medical Institute, Salk Institute for Biological Studies, La Jolla, California, United States of America

* magnus.nordborg@gmi.oeaw.ac.at

**Data Availability Statement:** All relevant data are within the manuscript and its Supporting Information files.

## Abstract

DNA cytosine methylation is an epigenetic mark associated with silencing of transposable elements (TEs) and heterochromatin formation. In plants, it occurs in three sequence contexts: CG, CHG, and CHH (where H is A, T, or C). The latter does not allow direct inheritance of methylation during DNA replication due to lack of symmetry, and methylation must therefore be re-established every cell generation. Genome-wide association studies (GWAS) have previously shown that *CMT2* and *NRPE1* are major determinants of genome-wide patterns of TE CHH methylation. Here we instead focus on CHH methylation of individual TEs and TE-families, allowing us to identify the pathways involved in CHH methylation simply from natural variation and confirm the associations by comparing them with mutant phenotypes. Methylation at TEs targeted by the RNA-directed DNA methylation (RdDM) pathway is unaffected by *CMT2* variation, but is strongly affected by variation at *NRPE1*, which is largely responsible for the longitudinal cline in this phenotype. In contrast, CMT2-targeted TEs are affected by both loci, which jointly explain 7.3% of the phenotypic variation (13.2% of total genetic effects). There is no longitudinal pattern for this phenotype, however, because the geographic patterns appear to compensate for each other in a pattern suggestive of stabilizing selection.

## Author summary

DNA methylation is a major component of transposon silencing, and essential for genomic integrity. Recent studies revealed large-scale geographic variation as well as the existence of major *trans*-acting polymorphisms that partly explained this variation. In this study, we re-analyze previously published data (The 1001 Epigenomes), focusing on CHH methylation patterns of individual TEs and TE families rather than on genome-wide averages (as was done in previous studies). GWAS of the patterns reveals the underlying

**Funding:** This work was funded in part by ERC AdvG 789037 EPICLINES to MN. The funders had no role in study design, data collection and analysis, decision to publish, or preparation of the manuscript.

**Competing interests:** The authors have declared that no competing interests exist.

regulatory networks, and allowed us to comprehensively characterize *trans*-regulation of CHH methylation and its role in the striking geographic pattern for this phenotype.

## Introduction

DNA cytosine-methylation (DNA methylation) is an epigenetic mark associated with diverse molecular functions, such as silencing of transposable elements (TEs) and heterochromatin formation. The majority of plant methylation is found in TEs, and there are three types of DNA methylation contexts: CG and CHG, both of which are symmetric, and CHH, which is not (H is A, T, or C). CG-methylation (mCG) and CHG-methylation (mCHG) can be maintained in a semi-conservative manner during DNA replication by *DNA METHYL-TRANSFERASE 1* (*MET1*) and *CHROMOMETHYLASE 3* (*CMT3*), respectively, whereas CHH methylation (mCHH) must be re-established every cell generation, presumably by one of two *de novo* pathways, one involving *CHROMOMETHYLASE 2* (*CMT2*), the other RNA-directed DNA methylation (RdDM) [1–3]. *CMT2* preferentially methylates heterochromatic non-CG cytosines [4, 5], while RdDM involves small RNAs that recruit *DOMAINS REAR-RANGED METHYLTRANSFERASE 2* (*DRM2*) to target regions throughout the genome [6, 7]. These pathways thus have separate target sites [4] and establish the genome-wide DNA methylation landscape in combination with maintenance and de-methylation pathways.

Natural variation for DNA methylation, superficially similar to DNA sequence polymorphism, is abundant in Arabidopsis [8, 9]. Although much of this variation likely reflects local sequence variation (e.g. segregating TE insertions), recent studies have revealed that a substantial part of the variation is controlled by *trans*-acting loci with genome-wide effects [10–12]. Understanding these *trans*-regulators is essential for understanding the genome-wide pattern of methylation variation, and could provide important clues to the function of DNA methylation.

The present study builds on previous results to comprehensively characterize *trans*-regulation of mCHH and its role in the striking geographic pattern for this phenotype. We achieve this by looking for genotype-phenotype associations at the level of individual TEs or TE families rather than genome-wide averages. As we shall see, this makes a huge difference.

## Results

### Major *trans*-regulators of mCHH levels

We first characterized average mCHH profiles of 303 TE families in each individual (S1 Table). Clustering analysis (based on the pattern across the 774 individuals) identified four groups, with the largest two roughly corresponding to the TE families that were previously shown to lose mCHH in RdDM and CMT2 pathway mutants (Fig. 1 [13]). The group corresponding to the RdDM pathway is mostly class I TEs and is enriched with RC/Helitron and DNA/MuDR, whereas the group corresponding to the CMT2 pathway is dominated by class II TEs and is enriched with LTR/Copia and LTR/Gypsy (note that targeting also strongly depends on element length and genome location; see S2 Fig).

GWAS for average mCHH levels of each TE family also identified the two main groups. Of 13 significant peaks (at FDR 20% and taking linkage disequilibrium (LD) into account; see Methods; Fig 1, S2 and S3 Tables), six are associated with the group corresponding to the RdDM pathway with strong signals at chr2:16719071 in the coding region of *NUCLEAR RNA POLYMERASE D1B* (*NRPE1*) as the largest component of RNA-polymerase V responsible for

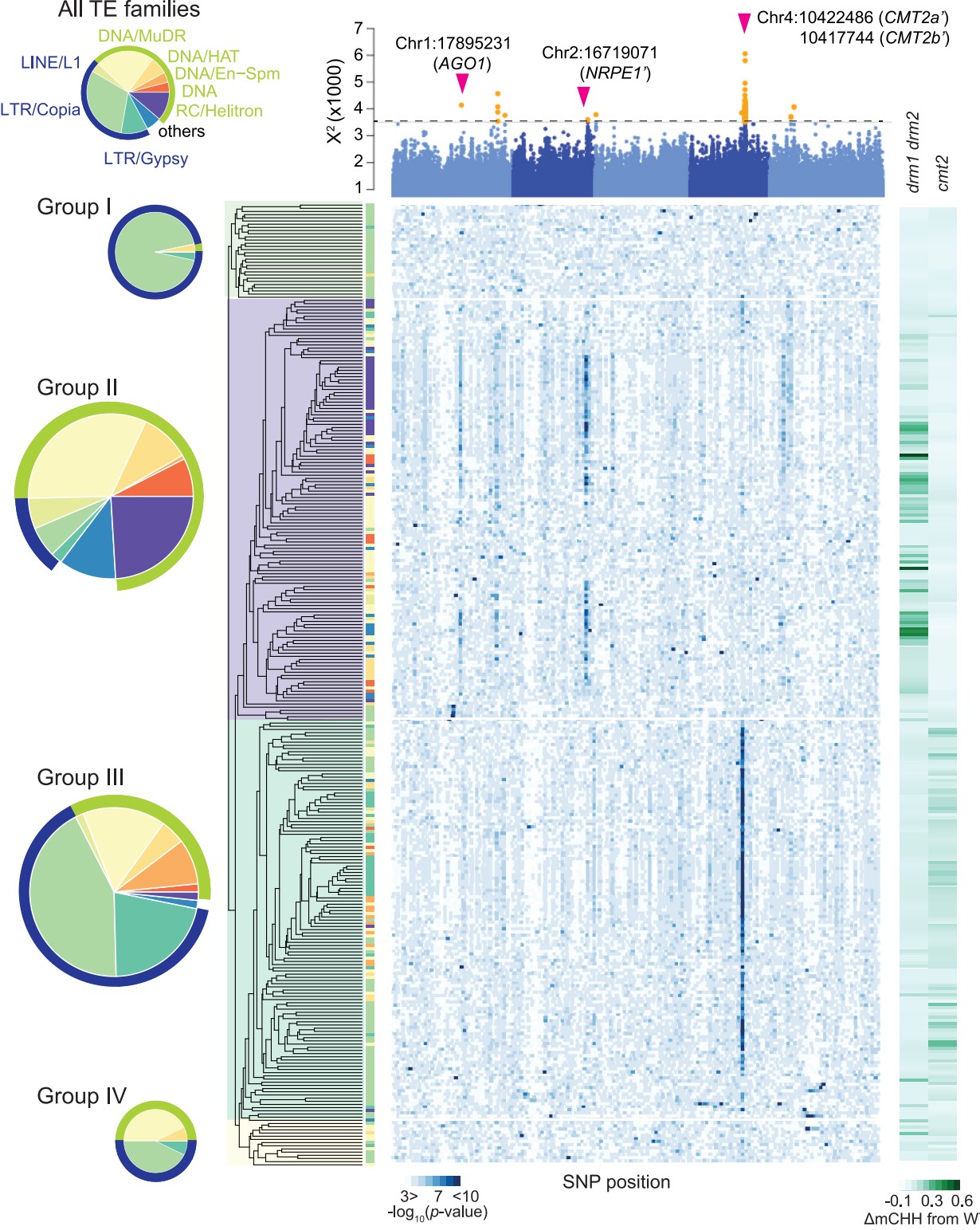

**Fig 1. The genetics of mCHH methylation at the level of TE families.** The heat map shows GWAS results for 303 TE families (each row is a family; columns indicate positions in genome; blue is more significant). The Manhattan plot on top shows integrated *p*-values from combining results across families (using $X^2$-statics). The horizontal line in the Manhattan plot gives FDR 20% threshold, with significant associations shown in yellow (see Methods, S1 Fig). Arrows indicate previously identified associations [11, 12] also identified here. TE-families (rows) have been

clustered based on average mCHH levels for 774 lines. The tip colors of the resulting tree correspond to TE superfamilies, and the superfamily composition for the four large clusters (Groups I-IV) is summarized by pie charts on the left. The greenish bars on the right show the reduction in mCHH levels of each TE family in *drm1 drm2* (RdDM pathway) and *cmt2* (CMT2 pathway) loss-of-function lines.

the RdDM pathway [7] and at chr1:17895231 in the promoter region of *ARGONAUTE1* (*AGO1*) recruits 21nt small RNA [14]. The remaining seven peaks are associated with the group corresponding to the CMT2 pathway with very strong signals at chr4:10417744 and chr4:10422486 in the coding or 3' region of *CHROMOMETHYLASE2* (*CMT2*) [5] (Fig 1).

The pattern of natural variation in mCHH is thus sufficient to outline pathways previously painstakingly discovered using traditional genetic screens, as well as to identify some of the major genes involved.

In addition to known genes, nine clear peaks suggest undescribed regulators of DNA methylation (S2 and S3 Tables). For example, the peak at chr1:27261944 is in the promoter region of a gene coding a DNAJ domain (At1g72416) that is a common component of DNA methylation reader complex [15], and the peak at chr4:9595111 is upstream of a histone H3K4-specific methyltransferase SET7/9 family gene (At4g17080) implicated in histone modification.

The four peaks that correspond to obvious *a priori* candidates are consistent with previous results [11, 12]. The peak near *AGO1* identifies the same top SNP as Kawakatsu et al [12] while the remaining three are in strong LD, but are much closer to the respective candidate genes, presumably because the present analysis, focusing on TE families rather than on average methylation levels, has higher resolution (Fig 2A). Thus chr2:16719071 is in LD with the previously identified chr2:16724013 [12], but is in the coding region of *NRPE1*, where it is LD with 12 non-synonymous polymorphisms and a three bp in-frame indel in the RNA polymerase domain. Similarly, chr4:10417744 is in LD with chr4:10454628 *CMT2b* (see [11]), but inside the coding region of *CMT2* and tagging two non-synonymous SNPs in the DNA methylase domain as well as a twelve base-pair deletion in the first exon. Finally, chr4:10422486 is in LD with chr4:10459127 CMT2a (see [11]), which is still outside the coding region, but presumably in the regulatory region.

For clarity, we will refer to the newly identified associations as *NRPE1'*, *CMT2b'* and *CMT2a'*. The non-reference *NRPE1'* allele is associated with decreased mCHH levels, whereas the non-reference alleles of *CMT2b'* and *CMT2a'* have negative and positive effects, respectively, in agreement with previous results [11].

GWAS for mCHH levels of 9,228 individual TEs that are present in all 774 lines showed a very similar pattern to GWAS for individual TE families. Although the *AGO1* peak was much weaker, the signals at *NRPE1'*, *CMT2b'*, and *CMT2a'* remain strong even at the level of individual TEs, with *NRPE1'* explaining 6.6% of the average mCHH variation on RdDM-targeted TEs, whereas the two *CMT2* alleles each explain about 4% (total 6.4%) of the variation on CMT2-targeted TEs (Fig 2B). Because the effect sizes are so large, and because the genes target different chromosomal regions (*NRPE1* mainly affects TEs in chromosome arms, whereas *CMT2* targets TEs in pericentromeric regions; see Fig 2B), these polymorphisms contribute substantially to shaping the genome-wide landscape of mCHH levels (Fig 2C), and the remainder of this paper will focus on them.

## Causality of *NRPE1* and *CMT2* alleles

Identifying the causal polymorphisms underlying a GWAS peak is notoriously difficult [17, 18]. However, because the phenotypes associated with the polymorphisms just described are so specific (multi-dimensional mCHH on hundreds or even thousands of specific TEs throughout the genome), it is possible to confirm the causal involvement of genes by comparison to mutant phenotypes. Specifically, we compared the estimated allelic effects of *NRPE1'*,

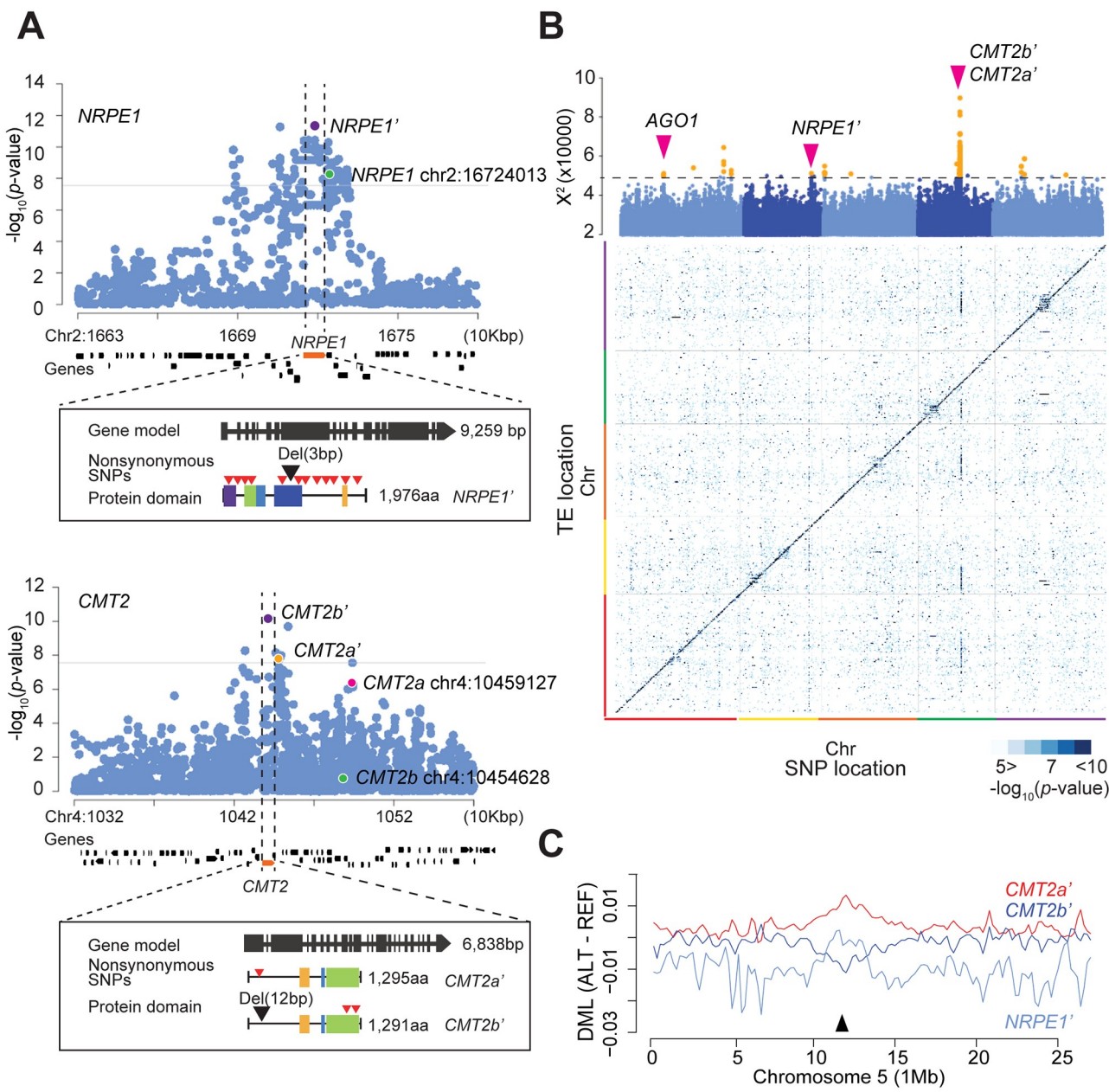

**Fig 2. *NRPE1* and *CMT2* are strong *trans*-regulators of mCHH levels.** (A) Examples of zoomed-in Manhattan plots for individual TEs targeted by *NRPE1* (AT3TE44975) and by *CMT2* (AT1TE41860). Horizontal lines show the 5% Bonferroni-corrected *p*-value threshold. Rectangles show gene models for the alleles identified [16]. Red and black triangles on the protein domain models indicate nonsynonymous SNPs and indels. (B) GWAS results for mCHH levels of 9,228 individual TEs in 774 lines (heat map) in each row with the integrated *p*-values by $X^2$ statics shown in the Manhattan plot above (yellow associations are significant using an FDR of 20%; see Methods). (C) Allelic effects on genome-wide mCHH levels (chromosome 5). Y-axis is the average differential mCHH levels between lines carrying alternative and reference alleles (300 Kbp sliding windows). The black arrow indicates the centromeric region.

*CMT2b'*, and *CMT2a'* on 9,228 TEs with the effects of knock-out mutations for 86 genes involved in gene-silencing, including *NRPE1* and *CMT2* [13]. The correlation between natural allelic effects and knock-out mutation effects for these genes was high, with the specific TEs significantly affected by the *NRPE1'* allele in GWAS also being affected by the *nrpe1-11* loss-

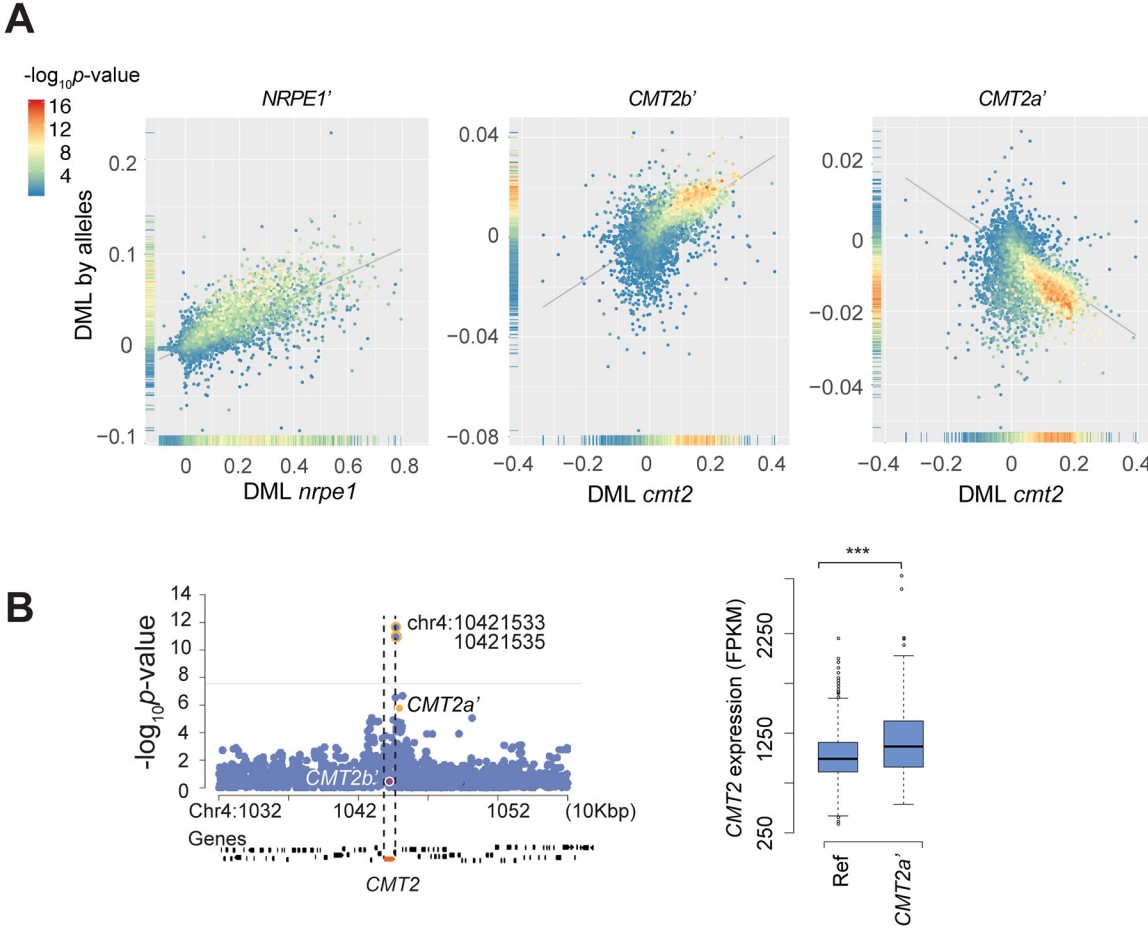

**Fig 3. The allelic effects of *NRPE1'*, *CMT2b'*, and *CMT2a'*.** (A) Comparison of the alleles to loss-of-function mutations of the corresponding genes. Note that *CMT2a'* increases mCHH levels relative to the reference allele. Scatter plots show correlations of differential mCHH levels (DML) induced by alleles and mutants for each TE. DML for alleles was estimated as average differences of mCHH levels between lines carrying reference and non-reference alleles, whereas for mutants it was estimated between wild-type and the *nrpe1-11* or *cmt2* loss-of-function. Colors of dots in the scatter plots show the significance of the allelic effects as -log₁₀p-value in GWAS. Density plots on Y and X-axis show distributions of the allelic effects for TEs. (B) Manhattan plots of *cis* peaks for *CMT2* expression (*n* = 665; leaf tissue under 21˚C) and effects of *CMT2* alleles. Horizontal lines show the threshold (*p*-value 5% Bonferroni correction), and identified SNPs in meta-analysis for mCHH variation of TE families were labeled (FDR < 20%). Boxplot shows *CMT2* expression of lines carrying reference or *CMT2a'* alleles. *** indicates *p*-value < 0.01 (Welch's *t*-test).

of-function allele, and TEs significantly affected by the *CMT2a'* and *CMT2b'* alleles in GWAS also being affected by the *cmt2* loss-of-function allele (Fig 3, S3 and S4 Figs).

Furthermore, the phenotypic correlation between *CMT2b'* and *cmt2* was much stronger than the correlation between *CMT2b'* and any other gene knockout (Fig 4), effectively confirming the causal role of *CMT2*—the alternative explanation would be that the identified non-synonymous polymorphisms in *CMT2* affect methylation via a closely linked unidentified gene that mimics the highly specific phenotypic effects of *CMT2* much better than any of the 85 other analyzed genes in these well-studied pathways. The correlation between the effects of *CMT2a'* and *cmt2* is notably weaker, perhaps because this allele affects expression like a moderate overexpressor (Figs 3B and 4). This may be worth exploring further.

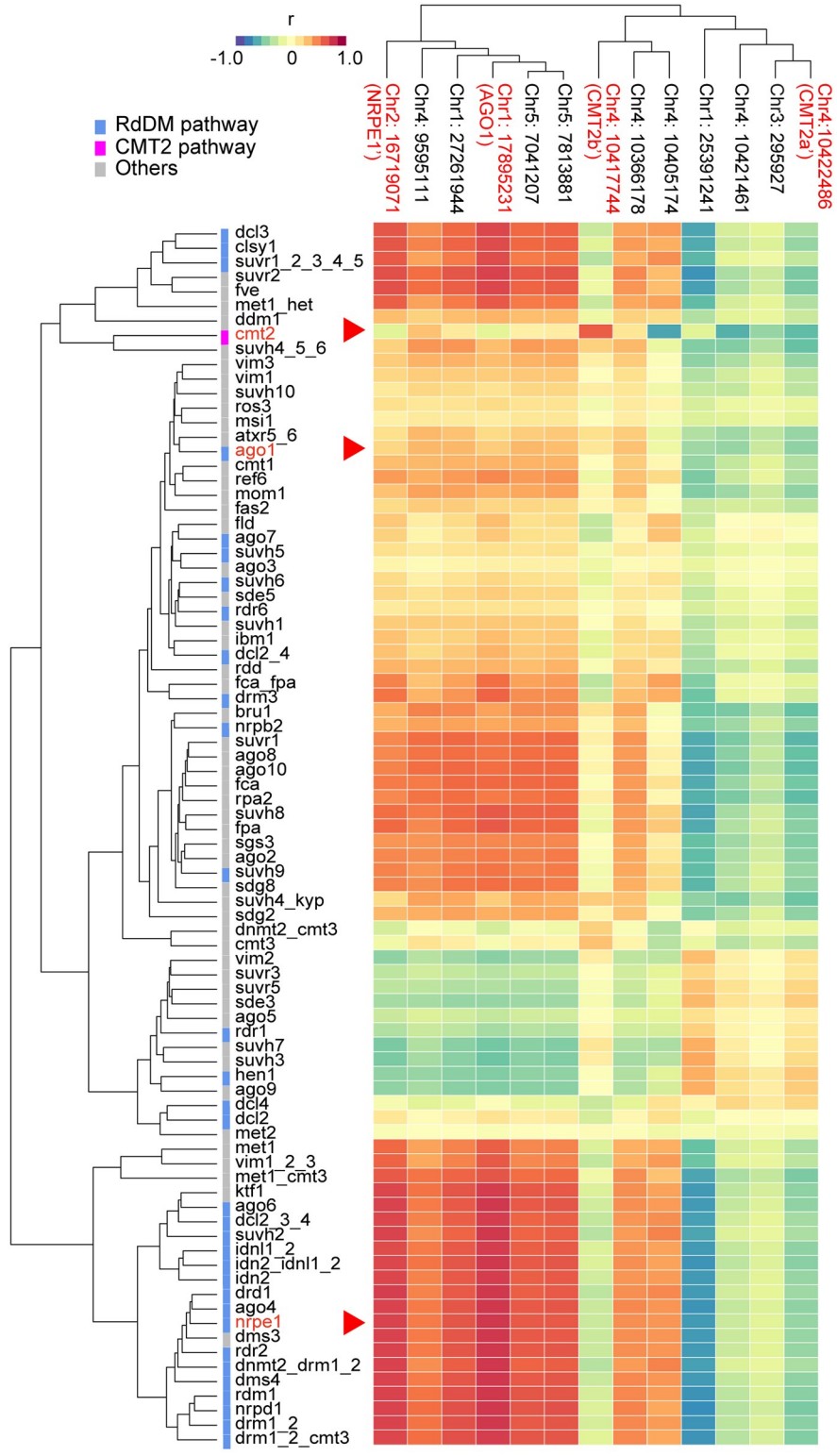

**Fig 4. Comparison of the effects of mCHH variation associated with 13 natural alleles to variation induced by knocking out 86 different genes involved in DNA methylation.** The heat map shows Spearman's correlation coefficients between SNP- and mutant-associated DMLs across 9,228 TEs. Both rows (mutants [13]) and columns (SNPs found in GWAS; see Fig 1, S2 and S3 Tables) have been clustered by similarity in DML pattern.

*NRPE1'*, by contrast, is clearly less specific, and showed strong correlations with loss-of-function phenotypes of nine genes in the RdDM pathway (including, of course, *NRPE1* itself). However, since none of these genes, nor any other plausible candidate, is located near *NRPE1* (S5 Fig), it seems reasonable to assume that the non-synonymous polymorphisms in this gene, particularly in RNA polymerase domain, cause a phenotype similar to knocking out *NRPE1* [19], rather than by somehow regulating an unknown member of the RdDM pathway (S5 and S6 Figs). The relative lack of specificity of *NRPE1* can also be seen from the comparison of natural alleles and knock-out mutations. Whereas variation at *CMT2* affects only a subset of TEs, variation at *NRPE1* affects all TEs, albeit to different extents (Fig 3).

In summary, we feel confident that both the *CMT2* and *NRPE1* alleles involve *cis*-acting polymorphisms that affect the phenotype via the corresponding genes. How this is done is of course not clear, but we note again that both *NRPE1'* and *CMT2b'* are associated with multiple non-synonymous SNPs, and that *CMT2a'* is associated with increased *CMT2* expression (Fig 3). Note that the same analysis does not work for the *AGO1* association, perhaps because the allelic effects are too small, or because the moderate *AGO1* mutant (*ago1-27* [20]) used in this analysis did not reflect the genuine effects on mCHH levels (S7 Fig).

## Apparent higher target specificity for natural alleles

The natural alleles thus show similar patterns to knock-out mutants, albeit with some notable differences. *CMT2b'* preferentially affects the same TEs as *cmt2* regardless of whether we consider the most significant or the largest effects (Figs 3 and 4, and S5 Fig). *CMT2a'* behaves similarly, but only when we consider the most significant effects, perhaps because this allele affects only *CMT2* expression. *NRPE1'* is more interesting, because while it is similar to the knock-out mutation in not affecting the LTR/Gypsy superfamily, it clearly affects the RC/Helitron superfamily preferentially, whereas *nrpe1-11* shows no such enrichment (S8A Fig).

This difference in specificity could be due to difference in target specificity between these alleles, but may also be explained by the population dynamics of TEs, because it turns out *nrpe1-11* strongly affects TE-superfamilies that have relatively low frequency in the population (like RathE3 cons and SINE; see S1 and S4 Tables, S8C Fig). These effects would be missed by the GWAS analysis of individual TEs, which only considers high-frequency insertions.

## *NRPE1'* allele broadly affects both the RdDM and CMT2-targeted regions

*CMT2* and *NRPE1* are considered to be parts of different pathways and target different TEs (Figs 1 and 2). However, as noted above, variation at *NRPE1* clearly affects methylation of CMT2-targeted TEs, whereas the converse is not true (Fig 3 and S9 Fig; *p*-value < 0.01).

We examined the joint allelic effect of *NRPE1'* and *CMT2b'* or *CMT2a'* on mCHH levels (Fig 5A). mCHH levels on the RdDM-targeted TEs are primarily decided by *NRPE1'*, and the effects of *CMT2b'* are insignificant (*t*-test $p = 0.58$ at center of RdDM-targeted TEs in all *CMT2b'* vs. *CMT2b'$_{ref}$* lines). The effect is qualitatively similar to the *cmt2* knock-out. On the other hand, *NRPE1'* additively suppresses mCHH levels of CMT2-targeted TEs ($p = 0.007$ at center of CMT2-targeted TEs in *NRPE1'* vs. *NRPE'$_{ref}$* lines), so that *CMT2b'/NRPE1'* (found in two lines: Lag1-5 and Bran-1) showed a 20% reduction of average mCHH levels relative to *CMT2$_{ref}$/NRPE1'$_{ref}$*. Although the genome-wide phenotypic variation explained by *NRPE1'* was not large (0.8%; see S3 Table), mCHH levels of CMT2-targeted TEs are well predicted by both loci (S9 Fig, S3 Table). The role of the RdDM pathway on the establishment of DNA methylation in CMT2-targeted TEs has been studied [4], and it appears to work on the edges of long TEs only (as shown in *cmt2*; see Fig 5A). In contrast, the effect of the natural allelic

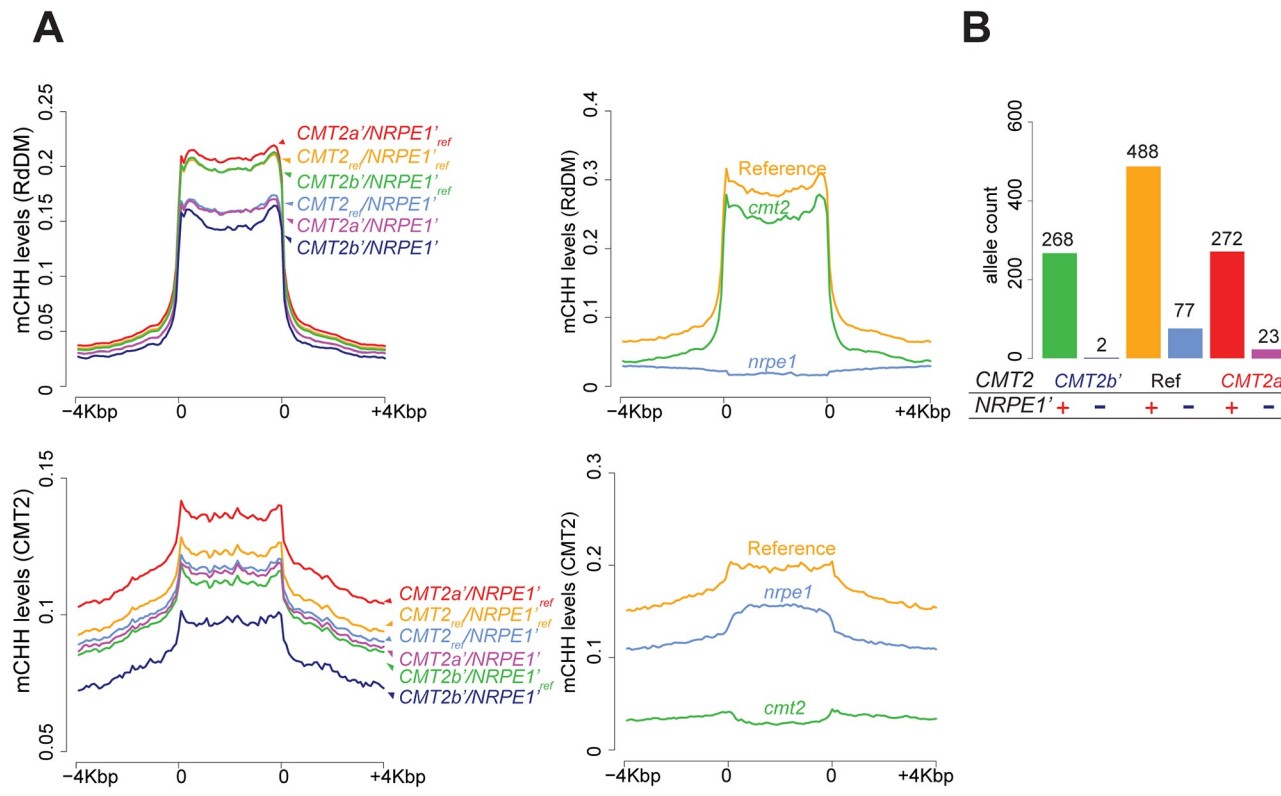

**Fig 5. *CMT2b*', and *CMT2a*' on mCHH levels in RdDM and CMT2-targeted TEs.** (A) mCHH levels of TEs for six genotypes (left) and *nrpe1-11* and *cmt2* (right). 5', TE body, and 3' regions were divided into 20 sliding bins for CMT2- and RdDM-targeted TEs. (B) Allele frequencies of combinational genotypes between *CMT2*', and *NRPE1*' in 1135 lines. *NRPE1*' '+' and '-' indicate reference and alternative alleles. Five lines carrying *CMT2b*'/*CMT2a*' were omitted.

variation at *NRPE1*' allele was observed over the entire TE, including the body. This suggests a qualitative difference between the natural alleles and the knock-out allele.

In summary, genotypes of *NRPE1* and *CMT2* generate further diversity of mCHH status over the genome. Given that both loci affect the pattern of methylation on CMT2-targeted TEs, it is worth noting that the allele frequencies at these two loci are strongly correlated. In particular, the genotype *CMT2b*'/*NRPE1*', which maximally suppresses mCHH levels is only found in 2 of 1135 lines—an order of magnitude fewer than expected under random mating, and significantly rare compared to genome-wide SNPs of identical frequency (Fig 5B and S10 Fig; *p*-value < 0.01). This suggests selection against this combination, perhaps to avoid genome-wide hypomethylation.

## *NRPE1* and *CMT2* alleles shape the longitudinal mCHH pattern

Previous studies have shown correlations of DNA methylation levels with several climate variables [10–12], but the genetic basis for this remains unclear. We examined whether the alleles at *NRPE1* and *CMT2* generate geographic patterns of mCHH levels (Fig 6). Variation at both loci show strong longitudinals patterns (*NRPE1*' $r^2$ = 0.37, *p*-value<2e-16; *CMT2b*' $r^2$ = 0.02, *p*-value = 4.4e-05; *CMT2a*' $r^2$ = 0.006, *p*-value = 0.03). At *NRPE1*, the alternative allele is essentially only found in the east, and this is the cause of a longitudinal cline in mCHH methylation on *NRPE1*-targeted TEs ($r^2$ = 0.024, *p*-value = 3.0e-05 vs $r^2$ = 0.002, *p*-value = 0.26 after regressing out *NRPE1*') even after correcting population structure (S11 Fig).

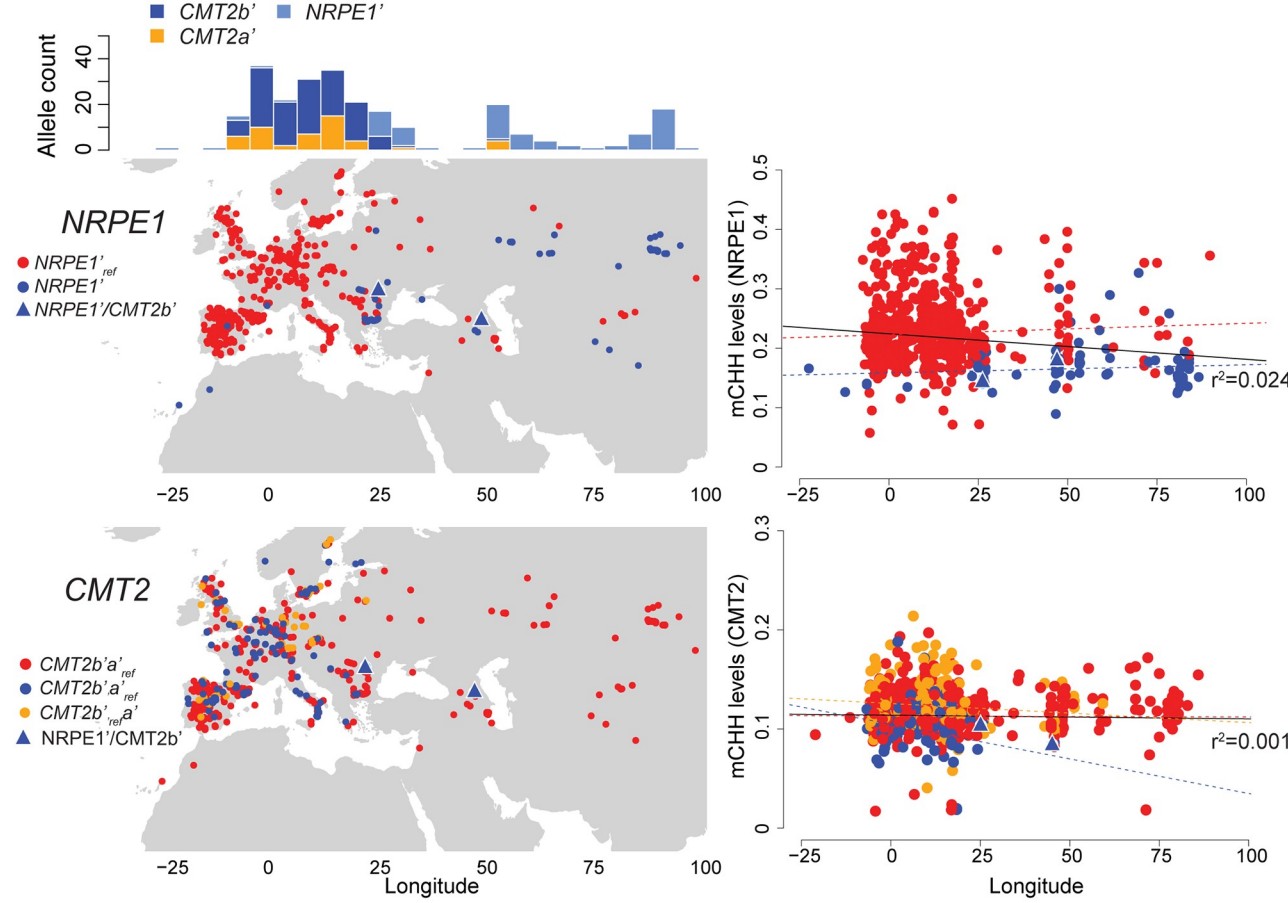

**Fig 6. Geographical distribution of *NRPE1* and *CMT2* alleles, and longitudinal mCHH variation.** Maps on the left show the distribution of *NRPE1'*, *CMT2b'*, and *CMT2a'* alleles, and the frequency of non-reference alleles along to longitude. Plots on the right show average mCHH levels of *NRPE1*- and *CMT2*-targeted TEs as a function of longitude. mCHH levels are average of *NRPE1'* and *CMT2b'*-targeted TEs. Colors of regression lines correspond to alleles; the black lines correspond to all lines.

At *CMT2*, both alternative alleles are limited to Europe, where they appear intermingled, but this causes no longitudinal cline for mCHH on CMT2-targeted TEs as the alleles have opposite effects relative to the reference allele ($r^2 = 0.001$; *p*-value = 0.39; Fig 6, S11 Fig). The distribution of *NRPE1'* alleles, which also affect CMT2-targeted TEs contributes to the lack of a longitudinal pattern (*p*-value = 0.03), consistent with the observation above that selection may be acting to stabilize methylation.

## Discussion

In this paper we re-analyze the 1001 epigenomes [12], focusing on mCHH patterns on individual TEs and TE families rather than on genome-wide averages performed in previous studies [10–12]. The advantages of this approach are evident. First, we were able to identify the well-known RdDM and CMT2 pathways using only natural variation data. This remarkable result is testament to the large effect of allelic variation in these pathways. We also identify several new associations, presumably corresponding to previously unknown members of these extensively-studied pathways (Figs 1 and 2). Second, the use of more fine-grained phenotypes allowed to refine previous associations, identifying candidate causal polymorphisms in both *CMT2* and *NRPE1* (Fig 2). Furthermore, by comparing the genome-wide mCHH pattern with

published data for loss-of-function mutations [13, 19], we were able to establish the causal involvement of these genes (Figs 3 and 4).

In terms of molecular mechanisms, our results largely confirm and complement previous studies [4, 5, 21]. The natural alleles of *CMT2* and *NRPE1* functionally behave much like loss-of-function alleles, albeit with some interesting differences that deserve further study. It is worth emphasizing in this context that these natural alleles have large effects, and are amenable to experimental studies. Perhaps because we are dealing with functional alleles, perhaps because we average over hundreds of lines, we get very clear pictures of which TEs are targeted by which *de novo* pathway (Fig 1 and S2 Fig). The mechanism underlying this targeting and the transition between pathways still remains unclear despite considerable effort.

Analysis of active TEs might be informative from this point of view. The current study is limited to TEs annotated in the reference genome, and present at high frequency in the population. New TE insertions are likely to generate DNA methylation diversity [22] but analysis of this will have to await long-read genome sequencing of many lines, which will let us capture rare insertions, and study *de novo* silencing. [12, 23, 24].

Finally, we confirm the existence of major *trans*-acting polymorphisms affecting CHH methylation [11, 12]. Based on currently available GWAS results, a genetic architecture characterized small numbers of genes of large effect is highly unusual, and is typically associated with adaptive polymorphism [25], but we can only speculate about what the adaptive value of variation in TE methylation would be. However, the idea of trade-offs and arms-races in a "genomic immune system" is not ridiculous—such mechanisms clearly maintain polymorphism in other defense systems [26]. The geographic pattern observed here, with linkage disequilibrium between unlinked loci (Fig 6), is certainly suggestive of selection.

## Materials and methods

### Methylation data

Bisulfite sequencing data, leaves of plants grown under ambient conditions at SALK, published in the 1001 epigenome project was mapped on each pseudogenome from the 1001 genome project [12, 27], using a Methylpy pipeline (https://bitbucket.org/schultzmattd/methylpy/wiki/Home). Methylation levels were calculated as weighted methylation levels [8]. TE regions were defined based on Col-0 by TAIR10 annotation, and 9,228 TEs having mapped reads in the region in all lines (*n* = 774) were used for all analyses as common TEs. CMT2 and RdDM-targeted TEs were defined as it having DML (>0.1) between wild-type and *drm1drm2* or *cmt2* in Col-0 [13] as previously described [12]. The classification of TE families and superfamilies was based on TAIR10 [28].

### Statistical analysis

**Clustering.** Clustering of TE families was conducted based on average mCHH levels across 774 lines (Fig 1). The values were transformed into rank order per line and analyzed by hclust function with R (https://www.r-project.org/), with the agglomerative method 'complete'. All other clustering analyses were conducted with raw values as described in results using hclust function with default settings.

**GWAS.** For GWAS of individual TEs and TE families, mCHH levels were transformed into rank order across lines. Average mCHH in TE families were calculated for it of common TEs. For GWAS of gene expression, 665 lines published in a part of the 1001 epigenome project were used [12]. We obtained normalized gene expression values using fragments per kilobase exon per million reads (FPKM) values published in Gene Expression Omnibus (GSE80744) and transformed it into the most normal by Box cox method. GWAS was

performed using a linear mixed model [29, 30] by LIMIX [31] with a full genome SNP matrix from the 1001 genome project (10,709,949 SNPs), and population structure was corrected by IBS matrix. A linear model without correction of population structure was conducted using lm function in R (https://www.r-project.org). SNPs that satisfied minor allele frequency (MAF) > 5% were used for association studies.

**Meta-analysis.** To combine *p*-values for each SNP calculated by GWAS, we used Fisher's methods as the following formula [32].

$$X^2 = -2\sum_{i=1}^{k} log(p_i) \tag{1}$$

where $p_i$ is *p*-value for ith GWAS, and *k* is the number of GWAS in the meta-analysis. $X^2$ follows $X^2$ distribution with $2k$ degrees of freedom. To optimize the threshold, we calculated false discovery rate (FDR) using the enrichment test with *a priori* gene list of 79 epigenetic regulators as described in [12]. The most significant *p*-value within 15 kb of a gene (MAF > 5%) was assigned as the significance of the gene.

LD ($r^2$) were calculated between all pairs of SNPs satisfied with the FDR threshold to determine independent GWAS peaks. In the case that a SNP pair has high LD ($r^2 > 0.2$), a SNP having lower $X^2$ scores was excluded from the list.

**Correlation of the allelic effects and molecular phenotypes.** Differential mCHH levels (DML) induced by alleles were estimated as differential average methylation levels between lines carrying the reference (Col-0) and the alternative allele for each TE. DML induced by mutants was calculated by the same way between wild-type and 86 loss-of-function mutants (Fig 4; GSE39901; [13]) and *nrpe1* mutants (GSE93558 [19]). Spearman's correlation coefficients were calculated between DML for natural alleles, and mutants and empirical *p*-values were estimated using permutation test with 1500 randomly picked up SNPs along to genome (S4 Fig).

**LD estimation.** D' as standardized linkage disequilibrium was calculated as D' = D/D$_{max}$ [33]. D' was calculated between the target SNPs (*NRPE1'*, *CMT2b'*, and *CMT2a'*) and genome-wide (unlinked) SNPs with same allele frequency of the target SNP. For example, *NRPE1'* (chr2: 16719071, MAF 9.0%) versus *CMT2b'* (chr4: 10417744, MAF 23.7%) was calculated between *NRPE1'* and all SNPs having the same MAF with *CMT2b'* (23.7%) on chromosome 1 and 3-5. The empirical *p*-value of observing an association was calculated using Fisher's exact test (one-sided).

**Analysis of geographic patterns.** Average mCHH levels of *NRPE1*- and CMT2-targeted TEs were calculated using TEs identified by GWAS (-log$_{10}$*p*-value > = 6 for *NRPE1'* and *CMT2b'*). Correlation between longitude and mCHH was calculated by a linear regression model for 728 lines ranging from longitude -25 to 100 in the 1001 epigenome project data (only SALK leaf samples).

## Supporting information

**S1 Fig. Enrichment of *a priori* DNA methylation responsible genes in meta-analysis.**
Enrichment and FDR 20% based on *a priori* genes (see Methods and also [12]). The horizontal dashed line at 0.2 corresponds to FDR 20%.
(PDF)

**S2 Fig. Effects of TE length and the location on target specificity of *NRPE1* and *CMT2*.**
Bar plots indicate the average length of TE families ordered by the length with GWAS *p*-values for three alleles (line plots; see also S1 Table) and the proportion of TEs located around

centromeric regions (black fraction in bar plots; 1Mbp from centromeric regions).
(PDF)

**S3 Fig. Effect of population structure on GWAS results.** (A) Scatter plots show correlations of differential mCHH levels (DML) induced by alleles and mutants for each TE. DML for alleles was estimated as average differences of mCHH levels between lines carrying reference and non-reference alleles, whereas for mutants it was estimated between wild-type and *nrpe1-11* or *cmt2*. Colors of dots in the scatter plots show the significance of the allelic effects as -$\log_{10}p$-value in GWAS (a linear model without correction of population structure). Density plots on Y and X-axis show distributions of the allelic effects for TEs. (B) Effects of population structure for mCHH levels of individual TEs. Scatter plots show -$\log_{10}p$-values estimated by a linear model (lm in X-axis) and a linear-mixed model (lmm in Y-axis).
(PDF)

**S4 Fig. Permutation tests for allelic effects.** Spearman's correlation coefficients (r) were calculated between DML of candidate mutants (*nrpe1-11* and *cmt2*) and 1500 randomly picked up SNPs over the genome (see Methods). Orange arrows show r of *NRPE1'*, *CMT2a'*, and *CMT2b'*. All allelic effects were significantly stronger than randomly picked up SNPs ($p$<0.001).
(PDF)

**S5 Fig. LD effects on the correlations between allelic effects and mutant phenotypes.** Each dot shows the absolute value of Spearman's correlation coefficients r between DML of the three alleles and 67 single knockout mutants [13] along with the gene location on the genome.
(PDF)

**S6 Fig. GWAS for *NRPE1* expression.** Manhattan plots and the *cis* peaks for *NRPE1* expression ($n$ = 665; leaf tissue under 21˚C). Horizontal lines show the threshold ($p$-value 5% Bonferroni correction).
(PDF)

**S7 Fig. The allelic effects of *AGO1* in the RdDM pathway and the similarity to *AGO1* activity.** Scatter plots show correlations of DML induced by *NRPE1'* and mutants, *nrpe1-11* and *ago1*, for each TE. DML for alleles was estimated as average differences of mCHH levels between lines carrying reference and non-reference alleles, whereas it for mutants was estimated between wild-type and *nrpe1-11* and *ago1* loss-of-function. Colors of dots in the scatter plots show the significance of the allelic effects as -$\log_{10}p$-value in GWAS. Density plots on Y and X-axis show distributions of the allelic effects for TEs.
(PDF)

**S8 Fig. Target specificities of the allelic effects of *NRPE1'*, *CMT2b'*, and *CMT2a'* on mCHH levels of individual TEs.** (A) Compositions of TE-superfamilies identified by GWAS, population-based average, or loss-of-function mutants at 0 to 90 percentile thresholds. (B) The scatter plot shows the correlation between DML induced by *NRPE1'* and *nrpe1* loss-of-function with 95% confident prediction. Blue dots indicate TEs showing *nrpe1-1* loss-of-function specific effects on DML, and red dots indicate TEs that were not detected by GWAS regardless of the DML (lm -$\log_{10}p$-value > 3). (C) Composition of TE-superfamilies shown in panel B (blue and red dots).
(PDF)

**S9 Fig. Allelic effects between RdDM and CMT2 pathways.** Correlation between molecular phenotypes of *nrpe1-11* and *cmt2* and the allelic effects on mCHH levels of TEs. *NRPE1*, *CMT2*-targeted, and untargeted TEs are shown in blue, red, and grey respectively based on

GWAS results (-log$_{10}$*p*-value>6 for *NRPE1*' and *CMT2b*'). Regression lines are corresponding to *NRPE1* and CMT2-targeted TEs.
(PDF)

**S10 Fig. Genome-wide pattern of LD for the *NRPE1* and *CMT2* alleles.** Plot A compares the value of D' between *NRPE1*' and *CMT2b*' (orange arrow) to the distribution of D' between *NRPE1*' and genome-wide (unlinked) SNP of the same frequency as *CMT2b*' on the left. The plot on the right shows the corresponding distribution of *p*-values calculated using Fisher's Exact Test (one-sided). The empirical *p*-value of observing an association this strong is less 0.01. Plots B and C show the same, focusing on *CMT2a*' and *NRPE1*', and *CMT2b*' and *NRPE1*', respectively.
(PDF)

**S11 Fig. Allelic effects on the geographical cline of mCHH levels.** Plots show average mCHH levels of *NRPE1*- and *CMT2*-targeted TEs by taking into account population structure (BLUP) as a function of longitude. mCHH levels are averages of *NRPE1*' and *CMT2b*'-targeted TEs. Colors of regression lines correspond to alleles; the black lines correspond to all lines.
(PDF)

**S1 Table. GWAS results for average mCHH of TE families.**
(XLSX)

**S2 Table. Top SNPs associated with mCHH variation (FDR20%).**
(PDF)

**S3 Table. Genetic effects on mCHH variation.**
(PDF)

**S4 Table. Compositions of TE superfamilies in Col-0 reference and it of common in the population (n = 774).**
(PDF)

## Acknowledgments

We thank Dr. Frederic Berger and Dr. Arturo Marí Ordóñez for critical reading of the manuscript, and Rahul Pisupati and Ümit Seren for technical support of data analyses (Gregor Mendel Institute of Molecular Plant Biology).

## Author Contributions

**Conceptualization:** Eriko Sasaki, Magnus Nordborg.

**Data curation:** Eriko Sasaki, Taiji Kawakatsu.

**Funding acquisition:** Joseph R. Ecker, Magnus Nordborg.

**Methodology:** Eriko Sasaki, Taiji Kawakatsu.

**Resources:** Taiji Kawakatsu, Joseph R. Ecker.

**Supervision:** Joseph R. Ecker, Magnus Nordborg.

**Visualization:** Eriko Sasaki.

**Writing – original draft:** Eriko Sasaki, Magnus Nordborg.

**Writing – review & editing:** Eriko Sasaki, Magnus Nordborg.

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
