## [Decision Letter · Decision Letter 0]

23 Nov 2019

Dear Dr Nordborg,

Thank you very much for submitting your Research Article entitled 'Common alleles of CMT2 and NRPE1 are major determinants of de novo DNA methylation variation in Arabidopsis thaliana' to PLOS Genetics. Your manuscript was fully evaluated at the editorial level and by independent peer reviewers. The reviewers appreciated the attention to an important topic but identified some aspects of the manuscript that should be improved.

We therefore ask you to modify the manuscript according to the review recommendations before we can consider your manuscript for acceptance. Your revisions should address the specific points made by each reviewer.

[LINK]

Yours sincerely,

Claudia Köhler

Associate Editor

PLOS Genetics

Gregory P. Copenhaver

Editor-in-Chief

PLOS Genetics

Reviewer's Responses to Questions

**Comments to the Authors:**

Reviewer #1: The study by Sasaki et al describes the identification of causal genes using GWAS for DNA methylation variation at transposon sequences within A. thaliana. Previous studies used bulk levels of DNA methylation within the 1135 epigenomes to identify putative natural variation at NRPE1 and CMT2. This study refines this approach by using mCHH at individual transposons, which significantly improve the resolution to pinpoint candidate natural variants. Numerous known and novel loci were identified and using previously published DNA methylome data from a collection of 86 A. thaliana mutants, they demonstrated the causal nature of the NRPE1 and CMT2 natural variants. These results further bolster this refined method for detection of natural variants controlling DNA methylation. Two examples of accessions possessing natural weak alleles of NRPE1 and CMT2 were identified at much lower frequency than expected by chance indicating selection against this combination. Lastly, integrating climate data there is an intriguing correlation between NRPE1 alleles and longitude.

Overall, this study nicely demonstrates the importance of the selection of phenotypic data for using GWAS to identify the causal nature of natural variants. The comments below are minor and are intended to improve this already strong manuscript.

1. The use of “de novo” methylation should be clarified. In the introduction it is stated that CMT2 is a de novo methyltransferase where as CMT3 is a maintenance methyltransferase. However, both recognize H3K9me2 to target DNA methylation. Given this mechanism is shared why is one considered maintenance and the other de novo. The type of methylation produce is irrelevant to this question. The use of CHH by the field to indicate de novo is overdue for a redefinition, as the Slotkin Lab has nicely shown, true de novo methylation is rare. Once true de novo methylation occurs at a region that previously was unmethylated all of the pathways, including RdDM ensue in maintenance methylation. This is important to this study due to the title and the opening introductory paragraph.

2. Line 13, CMT2 only binds H3K9me3 in vitro. As far as I know there is no known H3K9me3 in A. thaliana. And the one study that does report it is due to a antibody cross reaction with H3K36me3.

3. Figure 1—I think it is counterintuitive to display the methylation reduction the way that it is presented for drm1/drm2 and cmt2—it would be helpful if it was written under the key that you are showing percent reduction of methylation and what it is relative to.

4. Why is the pattern of correlation for differential CHH levels between cmt2/CMT2a’ and cmt2/CMT2b’ opposite?

5. Figure 5—Why is cmt2a’ / NRPE1ref methylation on cmt2 loci relatively high, but cmt2a’ / NRPE1’ and cmt2ref / NRPE1’ show about the same level of methylation on these loci

Reviewer #2: This manuscript presents a GWAS study of trans-regulation of Arabidopsis CHH methylation, based on 1001 Epigenomes data. It focuses on methylation of individual TEs and TE families, instead of genome averages (which were done by previous studies). The authors claim that this new analysis approach refined previous GWAS results and established causal relationship of the phenotype and CMT2 and NRPE1 alleles. In doing so, the authors also took advantage of the previously published methylome data from various Arabidopsis mutants and correlate with natural alleles. This analysis also identified several new associations; however, no further investigations were done. I think the results on CMT2 and NRPE1 are pretty convincing. I have a minor comment on Figure 5A. It is obvious that CMT2b’ is insignificant on mCHH levels of RdDM targets, while with NRPE1ref (orange and green lines basically overlap in Figure 5A top left panel and the authors also provided p value). However, it seems that CMT2b’ has lower methylation than CMT2ref or CMT2a’, while in NRPE1’ background (dark blue line vs. light blue and magenta lines). Moreover, although the authors say cmt2 knock-out is similar to CMT2b’, Figure 5A top right panel suggests that there is clear difference between reference and cmt2. In order to claim statistical significance or insignificance, p values of these pair-wise comparisons should be provided. In the case of AGO1, based on Figure S7, the effect of ago1 mutant on methylation is rather small. On line 125, did the authors mean to say the allelic effects are too small or the mutant effects are too small? The problem with this interpretation is that the ago1 mutant methylation data from Stroud et al Cell 2013 (which the authors of this study used according to Materials and Methods) is obtained from ago1-27, which is a hypomorphic, not a loss-of-function, allele (according to Morel et al Plant Cell 2002). True null or loss-of-function alleles of ago1 are lethal. In additional, there could be functional redundancy of other Agos that can compensate for reduction of ago1’s function in DNA methylation. Therefore, I would suggest the authors revise the part on ago1, or it will be misleading. Besides the concerns I raised above, the manuscript is well written and of interest to the plant epigenetic field.

**Have all data underlying the figures and results presented in the manuscript been provided?**

Reviewer #1: Yes

Reviewer #2: Yes

PLOS authors have the option to publish the peer review history of their article (what does this mean?). If published, this will include your full peer review and any attached files.

Reviewer #1: No

Reviewer #2: No

---

## [Editor Report · Decision Letter 1]

3 Dec 2019

Dear Magnus,

We are pleased to inform you that your manuscript entitled "Common alleles of CMT2 and NRPE1 are major determinants of CHH methylation variation in Arabidopsis thaliana" has been editorially accepted for publication in PLOS Genetics. Congratulations!

Yours sincerely,

Claudia Köhler

Associate Editor

PLOS Genetics

Gregory P. Copenhaver

Editor-in-Chief

PLOS Genetics

Comments from the reviewers (if applicable):

**Data Deposition**

http://datadryad.org/submit?journalID=pgenetics&manu=PGENETICS-D-19-01765R1

**Press Queries**

---

## [Editor Report · Acceptance letter]

13 Dec 2019

PGENETICS-D-19-01765R1 

Common alleles of *CMT2* and *NRPE1* are major determinants of CHH methylation variation in *Arabidopsis thaliana*

Dear Dr Nordborg, 

We are pleased to inform you that your manuscript entitled "Common alleles of *CMT2* and *NRPE1* are major determinants of CHH methylation variation in *Arabidopsis thaliana*" has been formally accepted for publication in PLOS Genetics! Your manuscript is now with our production department and you will be notified of the publication date in due course.

With kind regards,

Matt Lyles

PLOS Genetics

On behalf of:
